# Hepatoprotective Effect of Silver Nanoparticles at Two Different Particle Sizes: Comparative Study with and without Silymarin

**Mahmoud A. Elfaky** [1,2,*] **, Alaa Sirwi** [1] **, Sameh H. Ismail** [3] **, Heba H. Awad** [4] **and Sameh S. Gad** [4]

1 Department of Natural Products, Faculty of Pharmacy, King Abdulaziz University, Jeddah 21589, Saudi Arabia; asirwi@kau.edu.sa
2 Centre for Artificial Intelligence in Precision Medicines, King Abdulaziz University, Jeddah 21589, Saudi Arabia
3 Faculty of Nanotechnology for Postgraduate Studies, Sheikh Zayed Branch Campus, Cairo University, Sheikh Zayed City, Giza 12588, Egypt; drsameheltayer@yahoo.com
4 Department of Pharmacology, Faculty of Pharmacy, October University for Modern Sciences and Arts, 6 October City, Giza 12451, Egypt; hebahossam84@gmail.com (H.H.A.); sshaban@msa.edu.eg (S.S.G.)
* Correspondence: melfaky@kau.edu.sa

**Abstract:** Silver nanoparticles have been used for numerous therapeutic purposes because of their increased biodegradability and bioavailability, yet their toxicity remains questionable as they are known to interact easily with biological systems because of their small size. This study aimed to investigate and compare the effect of silver nanoparticles' particle size in terms of their potential hazard, as well as their potential protective effect in an LPS-induced hepatotoxicity model. Liver slices were obtained from Sprague Dawley adult male rats, and the thickness of the slices was optimized to 150 μm. Under regulated physiological circumstances, freshly cut liver slices were divided into six different groups; GP1: normal, GP2: LPS (control), GP3: LPS + AgNpL (positive control), GP4: LPS + silymarin (standard treatment), GP5: LPS + AgNpS + silymarin (treatment I), GP6: LPS + AgNpL + silymarin (treatment II). After 24 h of incubation, the plates were gently removed, and the supernatant and tissue homogenate were all collected and then subjected to the following biochemical parameters: Cox2, NO, IL-6, and TNF-α. The LPS elicited marked hepatic tissue injury manifested by elevated cytokines and proinflammatory markers. Both small silver nanoparticles and large silver nanoparticles efficiently attenuated LPS hepatotoxicity, mainly via preserving the cytokines' level and diminishing the inflammatory pathways. In conclusion, large silver nanoparticles exhibited effective hepatoprotective capabilities over small silver nanoparticles.

**Keywords:** silver nanoparticles; silymarin; cyclooxygenase enzyme 2; nitric oxide; interleukin-6; tumor necrosis factor alpha

## 1. Introduction

Nanoparticles are at the forefront of the rapidly evolving science of nanotechnology, and their unique size-dependent features make these materials superior and indispensable in a wide range of human activities [1]. These materials have emerged as key players in modern medicine in recent years, with therapeutic applications ranging from contrast agents in imaging to carriers for medication and gene delivery into malignancies [1–3]. Nanoparticles have a high surface area-to-volume ratio, allowing them to absorb significant amounts of medicines [4] and readily travel throughout the bloodstream and different tissues [5]. Their greater surface area provides them with distinct qualities, including improved mechanical, magnetic, optical, and catalytic properties, which increase their potential pharmaceutical utility as well as their potential toxicity [6,7]. Silver nanoparticles (AgNP) have the highest level of application among all used nanoparticles [5]. AgNP are widely used in medical applications due to their anti-microbial effect, anti-fungal action, and anti-viral activity [2]. Moreover, they play a crucial role in the modulation of cytokines to promote wound healing. Their use is not only limited to medical approaches, but

they can also be used for the disinfection of drinking water, anti-fouling for swimming pools, and as an anti-bacterial additive in water-based paints [8]. It is hypothesized that AgNP exert their anti-microbial effect via damaging the microbial membrane through the physicochemical attachment of AgNP on the cell's surface, leading to subsequent structural and functional alterations including gap formation, membrane destabilization, membrane piercing, and cytoplasm leakage [2]. Moreover, AgNP have the ability to cause sub-cellular structure damage, resulting in the release of free Ag+ ions and subsequent reactive oxygen species' generation, leading to the inactivation of proteins, enzymes, and nucleotides [9]. In previous studies, NP was proven to accumulate in tissues such as the spleen and liver, contributing to the nonspecific dispersion of nanotherapeutics to healthy organs [10]. It is believed that the increased cellular uptake and intracellular accumulation of AgNP in mammalian cells may add to their potential toxicity [11]. The surface area, particle size, surface charge, and zeta potential are critical for revealing mechanistic information in the uptake, persistence, and biological toxicity of NPs inside mammalian cells [9,12].

The liver is the principal organ involved in the metabolization, biotransformation, and detoxification of medicines and foreign chemicals, including nanoparticles [13]. Hepatotoxicity will cause complications such as splenomegaly, portal hypertension, hepatic encephalopathy, edema, and cancer of liver [14]. Some clinical reports indicate that AgNP could be used as a protective agent in hepatic injuries, but other reports showed that AgNP could lead to several hepatic complications [7]. AgNP have a stable structure with a lipid core of the particle size that enables them to reduce unwanted cellular uptake and prolonged drug release. AgNP can be used to target specific liver cells and deliver potent therapeutic action with low systemic toxicity. Understanding the potential toxicity of silver nanoparticles and the differential effect that nanoparticles of varying sizes may induce is critical [15]. The interaction of both nano and ionic forms of silver, with SH containing macromolecules like proteins due to the strong affinity of silver for sulphur, is considered an important mechanism for silver nanoparticles' toxicity [13,16,17]. AgNP, with a particle size from 1 to 100 nm, can cause apoptosis, oxidative stress, fibrosis, and elevation in ALT and AST, which indicate a hepatotoxic effect and appear in the form of necrosis and hepatocellular degeneration [18]. The mechanism of AgNP's toxicity remains unclear. Some researchers have proposed that toxicity occurs by AgNP as well as Ag+ ions. Other studies claimed that the AgNP may work as a "Trojan horse", releasing Ag+ ions after crossing the biological barriers, which are the main cause for cell injury [10]. On the other hand, silymarin, a hepatoprotective agent and bioactive compound of Silybum Marianum, was renowned for its high clinical efficacy and use in herbal treatment of chronic liver disease [19,20]. No health hazards are known in aggregation, along with properly managed compounds. Its hepatoprotective effect is mainly due to its antioxidant properties as well as its ability to promote the growth of new liver cells while also acting as a stimulant to the regeneration and detoxification of the liver [20,21]. Silymarin is used in the treatment of hepatitis and liver damage induced by excessive intake of alcohol. It works against oxidative stress, inflammatory responses, and benzoyl peroxide-induced fibrosis in mice model [21].

The liver slices' model used in this study was precision-cut liver slices, a stable and adaptable ex vivo model that preserves the hepatic environment's intricate and multicellular histoarchitecture. As such, they are an attractive model for studying the causes of liver injury and identifying potential treatment targets [22]. Liver slices are created using mechanical slicers such as the cryostat macrotome device, which makes reproducible slices with uniform thickness to enhance the optimum exchange of gases, nutrients, and waste. To maintain the slices' viability, they are incubated in culture for 1–10 days in a dynamic systems humidified incubator [23]. This is used to determine the mechanism of action and/or toxicity of certain drugs at a cellular and molecular level [24]. The model is considered to be consistent and rapid, as well as needing only a small number of animals, making it a cost-effective model. It can measure and analyze different samples at the same time. The in vitro model is stable for a long period of time, and it is easy to control the condition of the experiment [24]. Checking the viability of the liver slices in the ex-vivo

model is a crucial step. Several studies highligtened the importance of the quantification of the level of cytochrome p450 and albumin to ensure cell viability. Yet, some studies reported that although cytochrome P450 isoforms were strongly downregulated in the human cholestatic livers in vivo but were not regulated or somewhat upregulated in human PCLS [25]. Protein analysis of the slices is not prevalent in PCLS investigations [26]. One simple explanation is that most researchers pool numerous slices to collect enough protein, which requires many slices, yet it is mandatory if a time-course experiment or different conditions are being performed [26]. A number of previous studies were carried out with the same time of exposure as well as conditions [27,28]. Liver slices are the best in vitro method for showing the hepatotoxicity effect of the AgNP [29]. Lipopolysaccharide (LPS) is a component of Gram-negative bacteria that plays a significant role in its pathogenesis [30]. TLR4 is a receptor complex that is involved in the activation of the immune system by LPS. This is a vital activity that is important for the immune response due to Gram-negative bacteria and its endotoxin shock etiology [31]. The LPS model is most suited for analyzing influence on acute inflammation, because systemic effects are easily recognized and quantifiable. Furthermore, the LPS model is simple to use and is very replicable [32].

Our study aimed to investigate and compare the potential protective role of the AgNP using two different particle sizes and to assess the potential risk. AgNP of two sizes were employed in the treatment in this study: a small particle size (10 and 75 nm) and big particle size (250–300 nm) were included in the treatment of LPS liver slices, along with silymarin [27].

## 2. Materials and Methods

### 2.1. Silver Nanoparticles

We prepared the small particle size (AgNps) (10 and 75 nm) and large particle size (AgNpL) (250–300 nm). Chemical reduction produced AgNP from silver salt (AgNO3). This was accomplished by dissolving silver salt in water while reducing agents including ascorbate, sodium borohydride (NaBH4), hydrazine, citrate, and glucose were present. The AgNO3 solution (0.03 M) was dissolved in 200 mL of deionized water and heated to boiling, and then 0.3 M TSC was added drop by drop with careful stirring until the solution turned pale yellow. To eliminate light, the finished solution was chilled at ambient temperature in a dark, isolated environment [33,34].

### 2.2. Silver Nanoparticles' Characterization

The determination of the physiochemical properties of colloidal silver nanoparticles and their identification in addition to their index characteristic were done. Microscopic characterization was used to identify the morphology and surface topography, which were carried out by atomic force microscopy (AFM), scanning electron microscopy (SEM), and transmission electron microscopy (TEM). The morphology's identification of silver nanoparticles is very important, since significant changes in shape and size may greatly affect its biological activity. The identification was achieved using X-ray diffraction (XRD) and Raman to confirm the synthesis of colloidal silver nanoparticles without pollution from the synthesis process. The index was achieved to obtain information about its zeta potential and size using DLS (dynamic light scattering) [33].

#### 2.2.1. Identification Class

X-ray diffraction (XRD) measurements were performed using a Bruker D8 Discover system (Billerica, MA, USA). Cu K radiation was used as the X-ray source, with 32 mA current and 41 kV voltage. With 0.3°/min scan speed, the two angles varied from 35 to 90 degrees for silver nanoparticles. A Horiba lab RAM HR evolution spectrometer produced these Raman spectra. The laser line (532 nm edge) had a Raman shift range of 20 to 200 cm$^{-1}$ for the silver nanoparticles, a grating (450–850 nm), and a 10% ND filter to avoid oxidation of the silver nanoparticles. The acquisition period was 15 s, with four accumulations without a spike filter, and an X100 objective.

### 2.2.2. Index Class

The DLS and zeta potential analyzer were used to achieve DLS and zeta potential (Malvern, UK).

### 2.2.3. Microscopic Class

The surface topography of silver nanoparticles was determined using a 2D and 3D Atomic Force Microscope (AFM) (AFM 5600LS, Agilent, Santa Clara, CA, USA). To begin, the samples were treated with ultrasonic waves for 15 min at a frequency of 50 kHz, with an amplitude of 44 percent and 0.45 of a cycle (UP400S manufactured by Hielscher, Teltow, Germany). Finally, a thin layer was formed using a spin coater instrument, model Laurell-650 Sz, under vacuum at 700 rpm. Contact mode, Al tap, 0.71 In/S speed, I. gain 2, and P. gain 4 were used to create AFM pictures and data profiles at 200 nm × 200 nm and its zoom 100 nm × 100 nm. The surface morphology of selenium nanoparticles was studied using a scanning electron microscope (SEM) (JEOL, Akishima, Tokyo 196-8558, Japan). The silver nanoparticles were introduced to deionized water and sonicated for 15 min using an ultrasonic pump, with a 60 kHz amplitude of 41 percent and 0.41 of a cycle (UP400S manufactured by Hielscher, Teltow, Germany). TEM experiments were carried out using a TEM-2100 high-resolution electron microscope (JEOL, Akishima, Tokyo 196-8558, Japan).

### 2.3. Animals

Six adult male albino rats weighing between 200 and 250 g were used. The animals were kept at the MSA animal house in regular laboratory conditions and at a comfortable temperature. They were kept in plastic cages and fed conventional feed and water. The research was carried out in accordance with the MSA University's ethics committee rules and animal experimentation guidelines (code: PH8/EC8/2019F). The rats were slaughtered after reaching the desired weight (250 g) to obtain their livers.

The livers were extracted right away and stored at 4 °C for 24 h. Precision-cut liver slices were taken in the Animal Health Research Institute using a Cryostat macrotome apparatus, with the thickness of the slice optimized to 150 μm [23]. Under regulated physiological circumstances, the freshly cut liver slices (14 slices, each 150 m thick) were placed on a six-well plate [35]. A six-well dish was utilized to uniformly distribute the slices, with two slices in each well. All wells had the same weight and characterization [23].

The groups were created as follows: Group 1: (normal) was assigned to the six well plate; group 2: (control) LPS 0.5 mL (5 mg/13 mL distilled water) + Ringer solution; group 3: LPS + AgNP large particle size + Ringer solution; group 4: LPS + silymarin + Ringer solution; group 5: (treatment I) LPS 0.5 mL (5 mg/13 mL distilled water) + silymarin AgNP with small particle size 0.5 mL + 2 mL Ringer solution; and group 6: (treatment II) 0.5 mL LPS (5 mg/13 mL distilled water) + 0.5 mL silymarin AgNP (150–300 nm) + 2 mL Ringer solution. All wells contained liver slices and Ringer solution in an equal amount; the plates were then incubated in a humidified incubator in a research lab at MSA University under the conditions of 95% oxygen and 5% carbon dioxide to maintain the slices' viability. After 24 h of incubation, the plates were gently removed, and the supernatant and tissue homogenate were all collected and then subjected for analysis. These steps were repeated six times with each rat (N = 6).

### 2.4. Determination of IL-6, NO, and TNF-α Biomarkers

Hepatic inflammatory indices, including interleukin-6 (IL-6), were evaluated in the liver slices' homogenate using ELISA kits (MyBiosource, Inc., San Diego, CA, USA) according to the manufacturer's protocol. The rat's total nitric oxide (NO) was determined using an ELISA Kit (DEIA-BJ2206). The TNF-α were performed according to the Rat TNF-α ELISA Kit (Catalogue Number.CSB-E11987r).

*2.5. Determination of COX2 Enzyme by Quantitative Real-Time PCR*

2.5.1. RNA Extraction

The total RNA was extracted according to the manufacturer's instructions, using a Qiagen tissue extraction kit (Qiagen, Redwood City, CA, USA). For disruption and homogenization, 30 mg of the animal tissue sample was removed and deposited straight into a suitably sized vessel. The tissue was disturbed and lysed in RLT lysis Buffer, and the lysate was homogenized for 40 s in a tissue homogenizer. The supernatant was carefully collected and put into a new microcentrifuge tube after the lysate was centrifuged for 3 min at full speed. To the cleaned lysate, one volume (350 µL) of 70% ethanol was added. We transferred 700 µL of the sample to the RNeasy spin column and spun for 15 s at 8000 rpm in a 2 mL collecting tube. To wash the spin column membrane, 700 µL Buffer RW1 was added to the RNeasy spin column and centrifuged for 15 s at 8000 rpm. In a new 1.5 mL collection tube, we put the RNeasy spin column. To elute the RNA, 30–50 µL RNase-free water was poured directly to the spin column membrane and centrifuged for 1 min at 8000 rpm. For later usage, the eluate was transferred to a new Eppendorf tube and kept at −80 °C. Spectrophotometry was used to determine the purity (A260/A280 ratio) and concentration of RNA (dual wavelength Beckman Spectrophotometer, California, USA).

2.5.2. cDNA Synthesis

The total RNA (0.5–2 g) was utilized to convert to cDNA using a Fermentas' high-capacity cDNA reverse transcription kit. In the thermal cycler, 3 µL of random primers were introduced to 10 µL of RNA that had been denatured for five minutes at 65 °C. The temperature of the RNA primer combination was lowered to 4 °C. For each sample, the cDNA master mix was produced according to the kit's instructions and added to the sample (Table 1).

**Table 1.** Composition of cDNA master mix.

| Component | Volume |
|---|---|
| First strand buffer | 5 µL |
| 10 mM dNTPs | 2 µL |
| RNase inhibitor (40 U/µL) | 1 µL |
| MMLV-RT enzyme (50 U/µL) | 1 µL |
| DEPC-treated water | 10 µL |

A total of 19 µL of the master mix were added to the 31 µL of RNA–primer mixture, yielding a final concentration of 50 µL of cDNA. The combination was heated for one hour at 37 °C in a programmed thermal cycler and then inactivated for 10 min at 95 °C before being chilled at 4 °C. The RNA was converted to cDNA. We stored the transformed cDNA at –20 °C.

2.5.3. Real-Time qPCR Using SYBR Green I

An Applied Biosystem with software version 3.1 (StepOneTM, California, USA) was used to perform real-time qPCR amplification and analysis. At the annealing temperature, the qPCR test using the primer sets was optimized. Table 2 lists the primer sequences.

**Table 2.** The primer sequence of the studied gene.

| | Primer Sequence |
|---|---|
| COX2 | Forward primer 5′-GCAAATCCTTGCTGTTCCAATC-3′<br>Reverse primer 5′-GGAGAAGGCTTCCCAGCTTTTG-3′ |
| βeta actin | Forward primer 5′-TGTTTGAGACCTTCAACACC-3′<br>Reverse primer 5′-CGCTCATTGCCGATAGTGAT-3′ |

2.5.4. Preparation of the Reaction Master Mix for Q-PCR

For each sample, the following reagents and volumes, together with the running condition, are illustrated in Tables 3 and 4.

**Table 3.** Reagents and volumes added.

| PCR Reaction Mix Component | Volume |
|---|---|
| Forward primer | 1 μL |
| Reverse primer | 1 μL |
| SYBR green mix | 12.5 μL |
| cDNA template | 5 μL |
| RNase-free water | 5.5 μL |
| Total volume | 25 μL |

**Table 4.** Running condition for RT-PCR.

| Thermal Cycling Condition | | |
|---|---|---|
| Stage | Temp. | Time (s.) |
| Hold | 50 °C | 120 |
| | One cycle | |
| Denaturation | 95 °C | 15 |
| Annealing | 60 °C | 60 |
| Extension | 72 °C | 60 |
| | 40 cycles | |

2.5.5. Calculation of Relative Quantification (RQ) (Relative Expression)

The relative quantitation was calculated according to the Applied Biosystem software using the following equation:

$$\Delta Ct = Ct \text{ gene test} - Ct \text{ endogenous control}$$

$$\Delta\Delta Ct = \Delta Ct \text{ sample1} - \Delta Ct \text{ calibrator}$$

$$RQ = \text{Relative quantification} = 2^{-\Delta\Delta Ct}$$

The RQ is the fold change compared with the calibrator (untreated sample).

*2.6. Statistical Analysis*

All data were expressed as mean $\pm$ SD. The difference between groups was statistically analyzed by GraphPad Prism 6, using one-way analysis of variance followed by a Tukey–Kramer multiple comparison test. $p < 0.05$ was considered as significant.

**3. Result**

*3.1. Silver Nanoparticles' Characterization*

Identification class:

The XRD pattern for silver nanoparticles showed no characteristic peaks, while the Raman spectra of amorphous silver nanoparticles illustrated characteristic peaks at 52.3 and 146.5 of vibration. The silver nanoparticles unit cell peaks are shown in Figure 1.

Microscopic class:

The AFM images, as shown in Figure 2, illustrate the spherical shape of the silver nanoparticles, with a maximum height of 12 nm. The TEM image, as shown in Figure 3, also illustrates the spherical shape of the silver nanoparticles with a 15–25 nm size. The SEM images, as shown in Figure 4, illustrate the spherical aggregates of the silver nanoparticles.

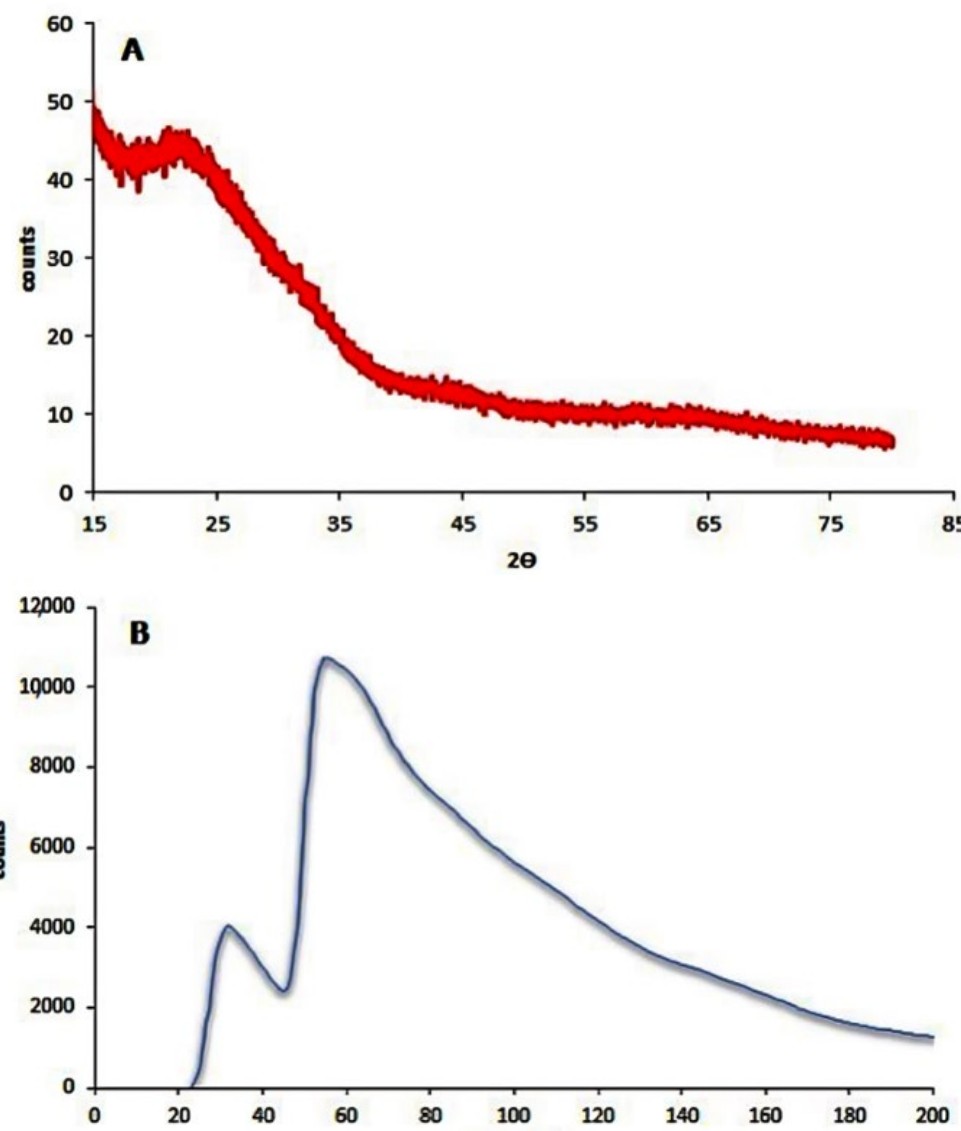

**Figure 1.** (**A**) XRD pattern of silver nanoparticles and (**B**) Raman spectra.

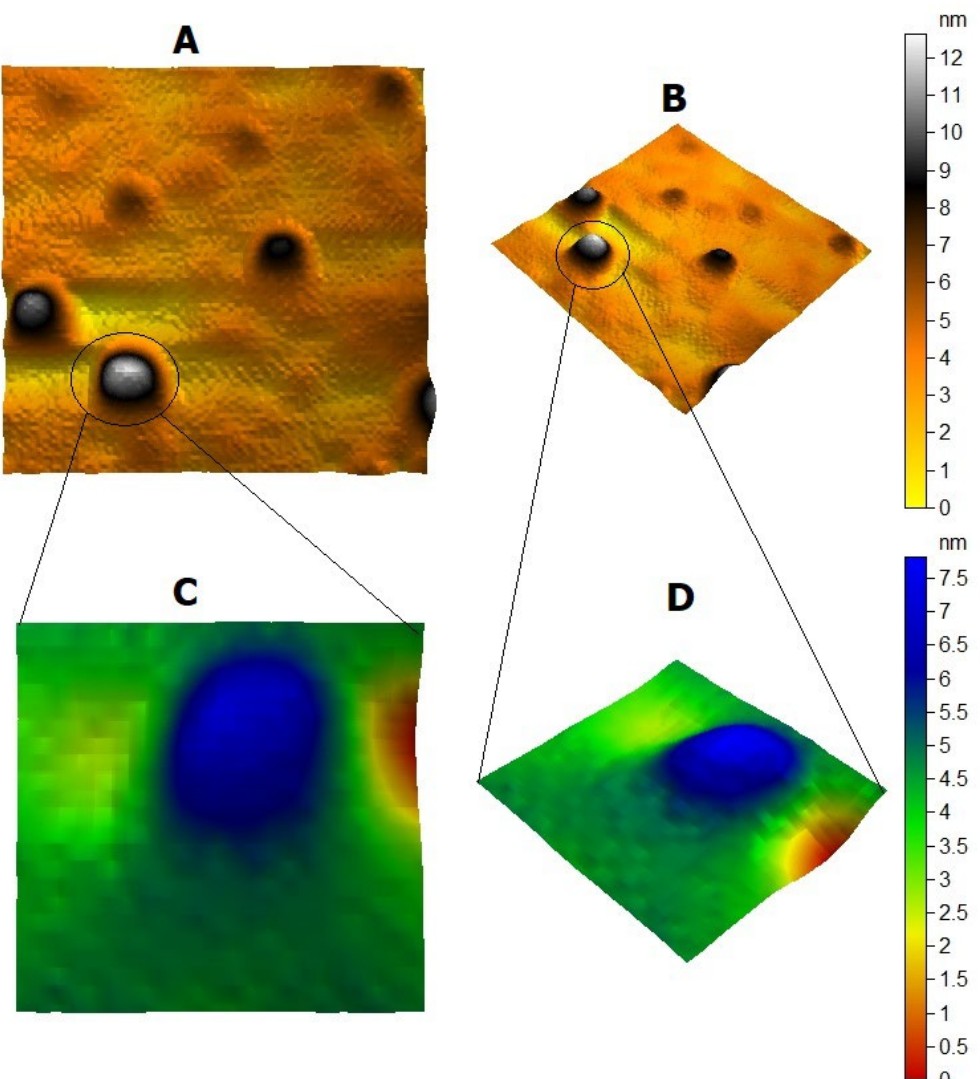

**Figure 2.** AFM image of colloidal silver nanoparticles. (**A**) 2D-view AFM image of 200 nm × 200 nm size. (**B**) 3D-view AFM image of 200 nm × 200 nm size. (**C**) 2D-view AFM image of 100 nm × 100 nm size. (**D**) 3D-view AFM image of 100 nm × 100 nm size.

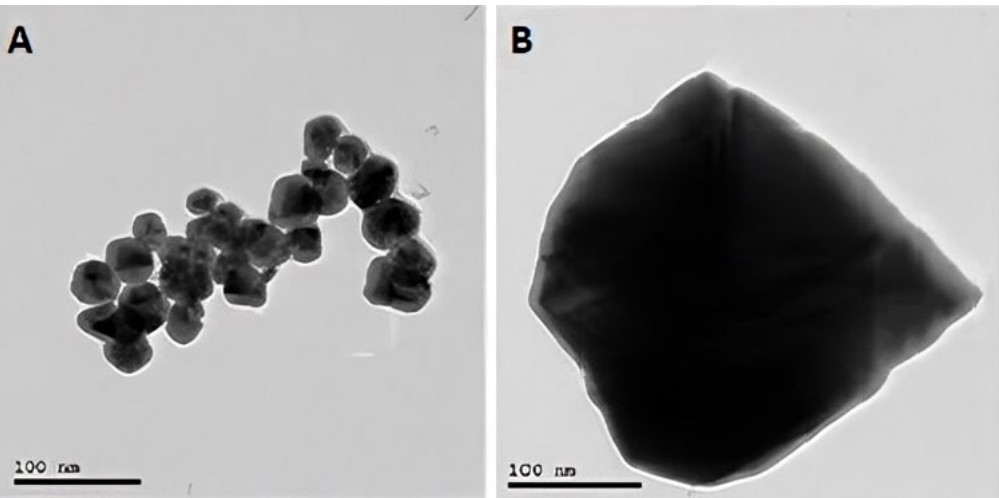

**Figure 3.** TEM image of silver nanoparticles (**A**) and bulk silver (**B**).

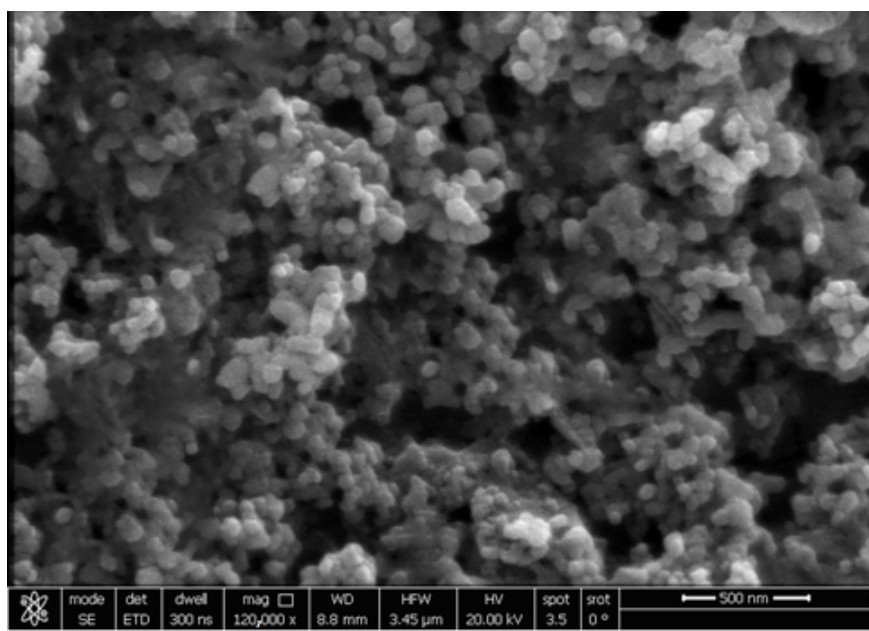

**Figure 4.** SEM image of silver nanoparticles.

Index class:
The zeta potential of the silver nanoparticles was –33.2 mV, while the size using DLS was 22 nm, as shown in Figure 5.

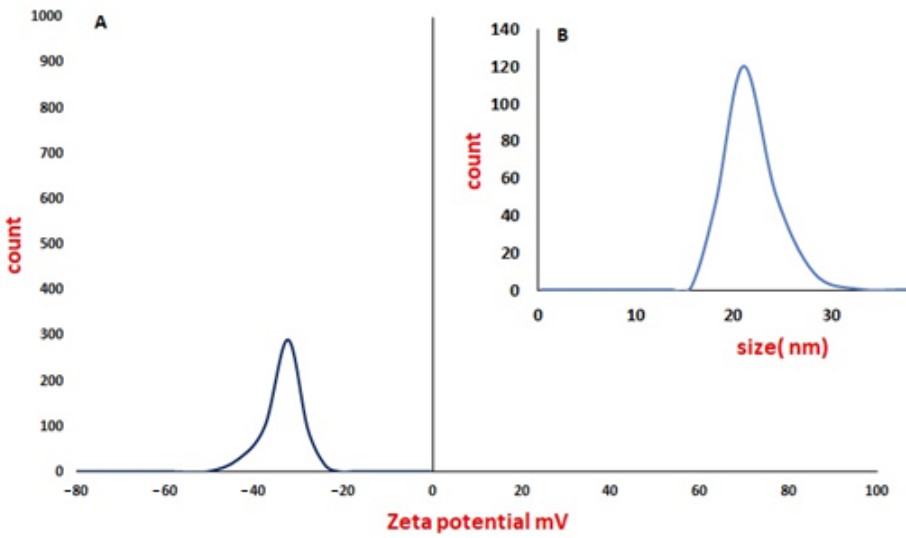

**Figure 5.** Zeta potential of silver nanoparticles (**A**) and DLS of silver nanoparticles (**B**).

The hepatocellular dysfunction and/or toxicity was significantly proven via the use of LPS in all the biochemical parameters measured; the levels of IL-6, No, TNF-$\alpha$, and cox-2 gene expression were 110.9 $\pm$ 0.21, 268.6 $\pm$ 0.31, 119.2 $\pm$ 0.47, and 135.3 $\pm$ 0.45, respectively, which was significantly higher than the normal levels of 37.57 $\pm$ 0.22, 34.5 $\pm$ 0.42, 24.33 $\pm$ 0.33, and 14.97 $\pm$ 0.35, respectively, at $p < 0.0001$ as shown in Figures 6–9.

*3.2. IL-6*

The LPS + silymarin-treated group showed a level of IL-6 that was significantly reduced to 60 $\pm$ 0.28 in comparison with the control group's level of 110.9 $\pm$ 0.21, while in the treated AgNpS group, the level was 39.85 $\pm$ 0.16, representing an extremely significantly decreased level in comparison with both the LPS + silymarin-treated group and control

group. In the group treated with AgNpL, the level was 41.30 ± 0.12, showing a significant decrease in comparison with the LPS + silymarin-treated group (Figure 6).

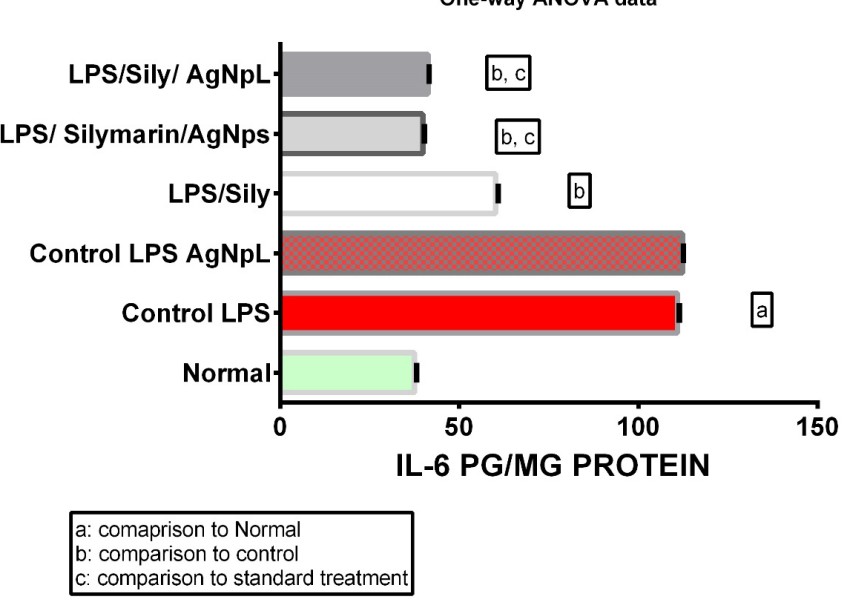

**Figure 6.** Expression levels of IL-6 among different groups.

### 3.3. NO

The levels of NO in the LPS + silymarin-treated group showed a significant reduction to 45.38 ± 0.29 in comparison with the control group's 268.8 ± 0.31. The treated AgNpS group showed a level of 58.02 ± 0.44, which reflected an extremely significant decreased level in comparison with both the LPS + silymarin-treated group and control group. In the group treated with AgNpL, the level was 38.87 ± 0.31, showing a significant decrease in comparison with not only the LPS + silymarin-treated group but also with the group treated with both silymarin and AgNpS (Figure 7).

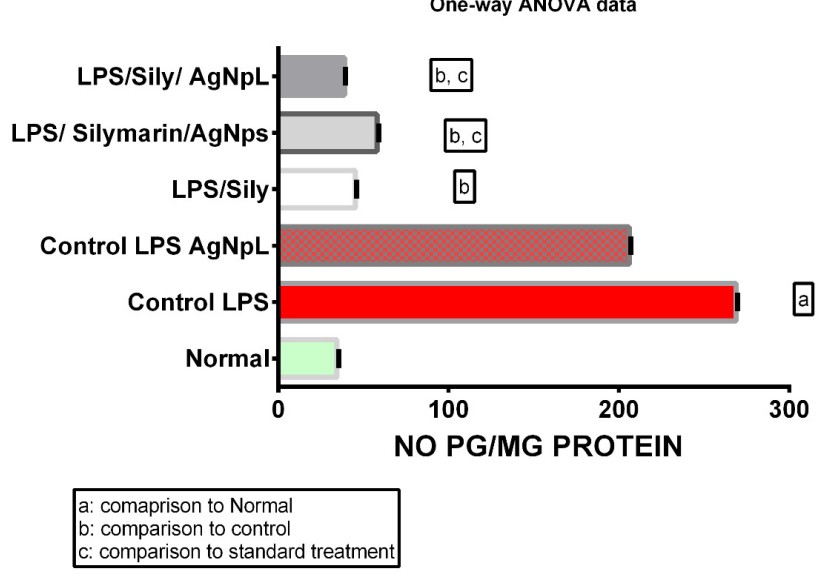

**Figure 7.** Expression levels of NO among different groups.

### 3.4. TNF-α

In the LPS + silymarin-treated group, the level of TNF-α revealed a significant reduction to 70 ± 0.36, in comparison with the control group's level of 119.2 ± 0.47. The treated

AgNpS group showed a level of 64.83 ± 0.47, which reflected an extremely significant decreased level in comparison with both with LPS + silymarin-treated group and control group. In the group treated with AgNpL, the level was 44.67 ± 0.49, showing a significant decrease in comparison with not only the LPS + silymarin-treated group but also with Group 5, which was treated with both silymarin and AgNpS (Figure 8).

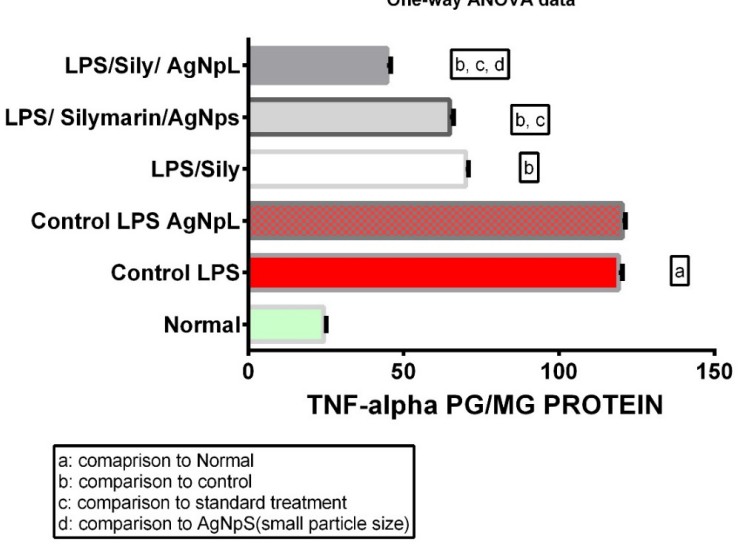

**Figure 8.** Expression levels of TNF-α among different groups.

### 3.5. COX-2 Gene Expression

The gene expression of COX-2 in the LPS + silymarin-treated group revealed a significant reduction to 127 ± 0.57, in comparison with the control group's level of 135.2 ± 0.45. The treated AgNpS group showed a level of 112 ± 0.0.36, which reflected an extremely significant decreased level in comparison with both the LPS + silymarin-treated group and control group. In the group treated with AgNpL, the level was 160.0 ± 0.57, showing a significant increase in comparison with the LPS + silymarin-treated group (Figure 9).

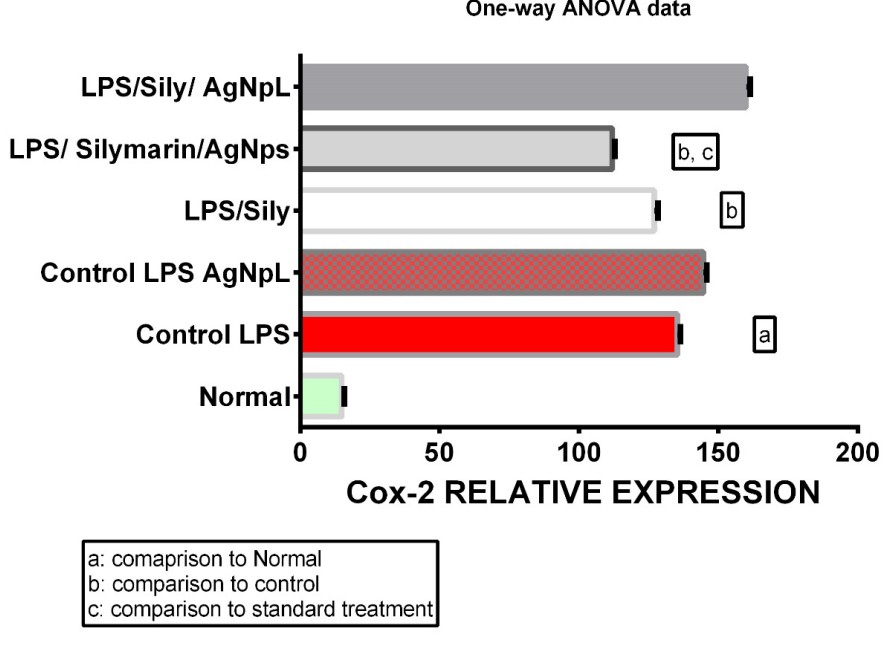

**Figure 9.** Expression levels of COX-2 among different groups.

## 4. Discussion

Recently, research has focused on the ability of nanoparticles to distribute and target drugs in order to explore new options for the treatment of several diseases [4]. The potential toxicity of AgNP has been extensively studied using in vitro and in vivo models [36]. It has been proven that the liver is the major accumulation site for AgNP [37]. Some clinical reports indicate that AgNP could be used as a protective agent in hepatic injuries [38], but other reports showed that AgNP could lead to several hepatic complications [39]. To explore the potential cytotoxic effect of AgNP, previous studies compared the effect of different particle sizes [40] and surface coatings [6] on the overall toxicity exerted by AgNP. Our study revealed the influence of small-size silver nanoparticles (AgNPs) and large-size silver nanoparticles (AgNPL) on liver tissue and the potential risk associated with the use of different NP sizes.

LPS is a component of Gram-negative bacteria that plays a significant role in its pathogenesis [30]. TLR4 is a receptor complex that is involved in the activation of the immune system by LPS. This is a vital activity that is important for the immune response, due to Gram-negative bacteria and its endotoxin shock etiology [31]. LPS can activate resident macrophages, resulting in cytokine release. Endotoxic shock is distinguished by the activation of nitric oxide synthase (NOS) in the liver. Inducible nitric oxide synthase (iNOS), which is activated by cytokines and LPS, produces massive amounts of nitric oxide (NO) [41]. NO is known to play a significant role in acute inflammation and sepsis. NO has anti-inflammatory and anti-endotoxemia effects on the liver in vivo. Furthermore, in response to endotoxin, inflammatory cells produce both pro-inflammatory and anti-inflammatory cytokines, such as interleukin-6 (IL-6) and the tumor necrosis factor-$\alpha$ (TNF-$\alpha$) [42]. LPS stimulates resident macrophages in the liver, resulting in the release of cytokines. These inflammatory reactions are generally seen as biliary obstructions and are associated in raised indicators of end-organ damage and death [43].

The tumor necrosis factor alpha (TNF-$\alpha$) is a multifunctional cytokine released by macrophage, lymphocytes, and natural killer cells, and it is linked with liver apoptosis. A study by Rimkunas et al. (2009) showed that TNF-$\alpha$ was linked with liver apoptosis, that the gene expression of TNF-$\alpha$ was upregulated in vitro at the lowest concentration used, and that gene expression of TNF-$\alpha$ in C3A cells decreased below the control group's gene expression as the silver concentration increased [41].

TNF-$\alpha$ and IL-6 are inflammatory mediators that contribute to the pathological complications observed in several hepatic diseases [43]. Furthermore, macrophages exposed to LPS showed a rapid and sustained expression of COX-2 mRNA, proving that LPS increases COX-2 gene transcription [45]. In this study, the LPS-induced inflammation was characterized by a significant elevation in IL-6, NO, and TNF-$\alpha$. These biochemical data reflected a severe LPS-induced inflammatory response that was further identified by the significant elevation in the COX-2 gene expression. These findings concur with the proven hypothesis that LPS induces deleterious inflammatory effects on liver slices [46–48].

On the other hand, silymarin, a milk thistle seed extract, has been used to treat hepatic disorders [20]. Silymarin is a free radical scavenger that also controls the enzymes involved in the development of cellular damage, fibrosis, and cirrhosis [49]. Silymarin has been shown to increase DNA and protein synthesis [50] and to inhibit NF-B, a transcription factor that regulates the expression of several genes involved in inflammation, cell defense, and cancer [15,51,52]. It has been shown that silymarin preserves intact liver cells or cells that are not yet permanently damaged by lowering oxidative stress and the resulting cytotoxicity, and thus it may be termed hepatoprotective [19]. In our study, treatment with silymarin prevented the increase of TNF-$\alpha$, which had a key role in increasing IL-6 and iNOS and in turn reduced the NO activation. In agreement with our findings, silymarin has been reported to attenuate the inflammatory response and suppress the activity of COX-2 induced by LPS in liver cells [53–56].

In all NP research, the composition, particle surface area, surface chemistry, and meticulous, accurate characterization of particle size and morphologic properties, particularly in

the physiological context, remain critical issues [57]. In this study, the XRD pattern of AgNP illustrated the amorphous nature of AgNP. Meanwhile, the Raman spectrum illustrated the fingerprint Raman shift pattern for AgNP. Moreover, the 2D and 3D AFM images and data showed the surface topography of AgNP. The AFM images illustrated its spherical shape with very sharp edges and homogenous size and shape. The results of the scanning electron microscope (SEM) and transmission electronic microscope (TEM) confirmed the AFM results. AgNP have a spherical shape, with their size ranging from 15 to 25 nm. The SEM image also determined the spherical shape of silver nanoparticles. Finally, the –33.2 mV zeta potential value illustrated the high stability of AgNP in an aqueous solution with colloidal properties. The DLS results illustrated the homogenous size with a size similar to its TEM measurement.

Several previous studies investigated the effect of nanoparticle size on cytotoxicity and cellular uptake [16]. Reports on the role of AgNP in regulating inflammatory response are controversial. Several studies have indicated that AgNP cause liver cell toxicity by triggering inflammation and eventually leading to different types of cell death [7]. It was proven that the relation between liver cell toxicity and AgNP is both time- and size-dependent [58]. One of the potential pathways reflecting the potential inflammatory effect of AgNP, especially those with a small particle size, is that they induce reactive oxygen species and free radicals, thus damaging the intracellular organelles and modulating the intracellular signaling pathways towards inflammation and apoptosis [7]. On the other hand, the large surface area of AgNP that release Ag+ ions are another crucial element that contributes to cytotoxic activity.

In our study, AgNP of two sizes were employed in the treatment in this study: a small particle size (10 and 75 nm) and big particle size (250–300 nm) were included in the treatment of LPS-contaminated liver slices, along with silymarin [27]. The biochemical results confirmed the anti-inflammatory effects of AgNP on both sizes, yet the results verified that the anti-inflammatory effects of AgNP were size-dependent. AgNPL significantly alleviated the effect of LPS on pro-inflammatory cytokines such as TNF-$\alpha$ and the pro-inflammatory mediator NO, compared with AgNps; however, for the pro-inflammatory cytokine IL-6, both the AgNPL and AgNps had the same significant lessening effect. These findings aligned with [27]. The difference in inflammatory mediator levels can be attributed to the different AgNP particle size used in our work. Since AgNps have a faster rate of silver ion (Ag+) dissolution due to their large surface area to volume ratio, AgNps have an enhanced distribution and toxicity of Ag compared with AgNPL [59,60]. This explains the significant elevation of NO and TNF-$\alpha$ levels in AgNps compared with AgNPL.

Hepatocytes respond both in vivo and in vitro to most of the stimuli that positively regulate COX-2 expression in other cell types, including lipopolysaccharide (LPS), interluekin l$\beta$ (IL-l$\beta$), tumor necrosis factor $\alpha$ (TNF-$\alpha$), and reactive oxygen intermediates. However, adult hepatocytes failed to induce COX-2 expression regardless of the pro-inflammatory factors used; only Kupffer cells and immortalized mouse liver cells retained the ability to express COX-2. COX-2 catalyzes the rate-limiting step in the synthesis of prostaglandins (PGs) and thromboxane, using arachidonic acid as substrate to generate PGH2. Interestingly, COX-2 expression in the liver protects against acute liver injuries by inhibiting apoptotic pathways in hepatocytes while increasing cell cycle progression and proliferation. COX-2 expression in the liver causes an increase in anti-apoptotic genes as well as the activation of cell survival proteins, such as phospho-Akt and phospho-AMP-kinase, following injury. COX-2 inhibition, on the other hand, eliminates these protective benefits. Our results showed a significant increase in COX-2 expression in the AgNPL group compared with the AgNps group. Our study confirmed the findings of the aforementioned studies and introduced the notable role of AgNPLs.

## 5. Conclusions

Hepatic injury is one of the most serious conditions that affect human lives. Hepatotoxicity can be caused by several factors, such as alcohol intake, viruses, autoimmune hepatitis,

non-alcoholic fatty liver diseases, inherited diseases that affect the liver, and chemically or drug-induced hepatic injury, such as silver nanoparticles with a small particle size.

In our study, the effect of silver nanoparticles with small and large particle sizes on the liver were thoroughly investigated. We clearly noticed that same pharmacological response could be obtained using large-sized particles, which could be a safe step to ceasing the use of the small-size silver nanoparticle, which are known to be more toxic and highly precipitated in tissues.

**Author Contributions:** Resources, supervision, validation, and writing—review & editing: M.A.E. and H.H.A.; methodology, visualization, writing—original draft, and project administration: A.S.; formal analysis, investigation, methodology, data curation, and software: S.S.G. and S.H.I. All authors have read and agreed to the published version of the manuscript.

**Funding:** This project was funded by the Deanship of Scientific Research (DSR) at King Abdulaziz University, Jeddah, under grant no. G: 138–166–1441. The authors, therefore, acknowledge with thanks DSR for technical and financial support.

**Institutional Review Board Statement:** The study protocol was approved by the ethics committee of the Faculty of Pharmacy, MSA University, Ethics Committee (code: PH8/EC8/2019F).

**Informed Consent Statement:** Not applicable.

**Data Availability Statement:** Not applicable.

**Acknowledgments:** This project was funded by the Deanship of Scientific Research (DSR) at King Abdulaziz University, Jeddah, under grant no. G: 138–166–1441. The authors, therefore, acknowledge with thanks DSR for technical and financial support.

**Conflicts of Interest:** The authors declare no conflict of interest.

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
