# Peer review of "Hepatoprotective Effect of Silver Nanoparticles at Two Different Particle Sizes: Comparative Study with and without Silymarin"

_cimb, doi:10.3390/cimb44070202_

Round 1
Reviewer 1 Report
I found the manuscript " Hepatoprotective effect of Silver nanoparticles at two different 2
particle sizes: Comparative study with and without Silymarin." is interesting, which is the focuses on reveal the mechanism of silver nanoparticles in two levels: protection 15 and/or toxicity with respect to the usage of two different particle size of silver nanoparticles The topic is of current interest and suited for the journal; anyways, some minor modifications of the submitted paper are recommended before publication.
Concerns are mentrioned bellow.
Author should re write the abstract with clear objective.
In introduction section need to revise and add some recent references.
Author should add ethical sentence according to the journal (Bottom of the manuscript).
Author may write comparison between silver nanoparticle and biological synthesis nanoparticles specially toxicity and efficacy.
Why twi different size, is ther any mnechanism if yes plaz discuss in the discussion section more with new references.
Figure 1 replace with high resulpution figure.
Discussion:
Kindly re write
XRD pattern of Silver ………………………… tern for silver nanoparticles
Author may cite bellow mentioned reference
Oves M, Ahmar Rauf M, Aslam M, Qari HA, Sonbol H, Ahmad I, Sarwar Zaman G, Saeed M. Green synthesis of silver nanoparticles by Conocarpus Lancifolius plant extract and their antimicrobial and anticancer activities. Saudi J Biol Sci. 2022 Jan;29(1):460-471. doi: 10.1016/j.sjbs.2021.09.007. Epub 2021 Sep 13. PMID: 35002442; PMCID: PMC8716933.
Reviewer 2 Report
A brief summary
The article by Elfaky et al presents a study of the effects of the exposure to AuNPs on slices of liver of rat. They produce and characteraized one AuNP that they exposed to slices of liver. The effect of a single condition exposure (dose and time) was analyzed by the quantification of some inflammatory cytokines (IL-6, TNF) or mediators (NO) and Cox-2 gene. Control molecules ( LPS, Silymarin) were used in the exposure assays.
The declared aim of the paper was to support the benefit of the slices of liver model to monitor hepatic toxic effects of products taking AuNP as example.
The article presents too many flaws in the design of the experiments, in the produced data and in the conclusions to provide valuable contribution. The slices of liver is already widely used and the results did not demonstrate news.
Broad comments
The title and the abstract mention the study of “ two different particle size of silver nanoparticles” where the article present only data with a single AuNPs (32 nm of size). Therefore, the title and the abstract do not correspond to the core of the article.
The method used for statistics must be explained in the Methods section aswellas the number of independent experiments.
The results about “Silver nanoparticles characterization” must be more detailed in providing the conclusions about the characterization of the AuNP by each physical methods. Text must be in accordance with the data presented in the figure; for instance (line 247) it is written “ the size using DLS was 32nm 247 as shown in figure 4” wheras the DLS presented in figure 5 (not 4) is about 20 nm obviously.
A table combining all physical data may help.
The use of LPS in the functional assays is not sufficiently justified. Why LPS is used as toxicity control and why chemical toxics were not? In the same idea, the read out of the toxicity includes IL-6, TNF, cox-2 gene, NO without a clear justification.
The presentation of the IL-6 data does not fit with the figure 6. What is the “ Standard treated group » (line 272) and where is it in figure 6. The groups in the figure and in the text must be definitively defined. The statistics data have to be clarified (p value or else accordingly to the methods). The same remarks for NO and TNF data.
The conditions of the exposure to AuNPs are not indicated; apparently, relies on a single dose of AuNP and on single time of exposure. Taking into account the parameters, which have been recorded, have different kinetics (secetion of cytokines versus transcription of gens) this set up appears inappropriate.
There is no indication on the survival f of the cells. Thus, a decrease of activity could be due to either a direct toxicity effect impacting the viability and/or to an effect specific to the parameters analysed.
The discussion section needs to deeply modified. The discussion is not the place to repeat the list of methods, or to present the data as it stands yet. This has to be moved to the results section. Instead, it would be interesting to get more words on the comparison between the data generated on using “liver slices” and those using other methods like cell culture in vitro or exposure in vivo.
Specific comments
The statement of AgNPs as “ protective agent “ or could “ lead to several hepatic complications” should be supported by references (line 40-41);
Correct
Line 35 “…, It also help…” by “…, it also helps…”
Line 39 « that’s may leads » by « that may lead”
Line 42 is not correct, missing a verb.
Line 47 the statement “have a size in the Range of 10-1000 nm” should be revisited since it is when the size is below 200 nm that the NP qualification is used.
Line 49 “ This done by dissolve silver “ is not correct, missing a verb.
Line 50-51 “ The size and shape of silver nanoparticles influences catalytic, optical” correct “ “ The size and shape of silver nanoparticles influence catalytic, optical”
Line 65 “ that toxicity occur by both AgNPs” correct by “ that toxicity occurs by both AgNPs”
and so on
Line 162: provide correct citation (18) in the text for “ Olinga and Sshuppan (2013)”
There are too many mistakes in the English to be all listed here. Furthermore, editing must be revised (erratic use of capitals in the middle of sentences, wrong punctuation, sentences lacking verb etc).
Check the citation of figure I the text, for instance line 242 “ AFM images as shown in figure 7 illustrate” acutally AFM is presented in figure 2 !
In the references
Line 417 correct the doi, should be “doi:10.1007/S00204-021-02992-7”
The reference de Graaf IA et al 2010 can be added for liver slices methods, should be valuable for interested researchers (de Graaf IA, Olinga P, de Jager MH, Merema MT, de Kanter R, van de Kerkhof EG, Groothuis GM. Nat Protoc. 2010 Sep;5(9):1540-51. doi: 10.1038/nprot.2010.111).
Round 2
Reviewer 1 Report
Accept in current form
Author Response
The reviewer recommends acceptance in current form
Reviewer 2 Report
The revised version of the article by Elfaky et al answers correctly to my remarks. The discussion was deeply revised, thus providing now an interesting version.
Minor points
Legend Figures 6-9, in the boxes: correct “a:significant in comparison …” by “…. comparison …”
Author Response
Legend of Figures 6-9 have been corrected as per reviewer's suggestion